# Spatial-Temporal Context Model for Remote Sensing Imagery Compression

## ABSTRACT

With the increasing spatial and temporal resolutions of obtained remote sensing (RS) images, effective compression becomes critical for storage, transmission, and large-scale in-memory processing. Although image compression methods achieve a series of breakthroughs for daily images, a straightforward application of these methods to RS domain underutilizes the properties of the RS images, such as content duplication, homogeneity, and temporal redundancy. This paper proposes a Spatial-Temporal Context model (STCM) for RS image compression, jointly leveraging context from a broader spatial scope and across different temporal images. Specifically, we propose a stacked diagonal masked module to expand the contextual reference scope, which is stackable and maintains its parallel capability. Furthermore, we propose spatial-temporal contextual adaptive coding to enable the entropy estimation to reference context across different temporal RS images at the same geographic location. Experiments show that our method outperforms previous state-of-the-art compression methods on rate-distortion (RD) performance. For downstream tasks validation, our method reduces the bitrate by 52 times for single temporal images in the scene classification task while maintaining accuracy.

## CCS CONCEPTS

• **Computing methodologies → Image compression**.

## KEYWORDS

Remote Sensing Imagery, Compression, Context Model

## 1 INTRODUCTION

Remote sensing (RS) images play an important role in resource surveying, urban planning, agricultural development, national security, and other related areas. With the continuous advancement of satellite RS imaging and processing technologies, we can acquire a large number of RS images with various spatial, temporal, and spectral resolutions. For example, the total volume of RS images produced by satellites can reach the petabyte (PB) scale in a single day. However, managing, transmitting, and using these vast amounts of RS images requires significant energy consumption, making it challenging for users to access and store these data. Therefore, exploring effective compression methods for RS imagery is vital for sustainable development and reducing data acquisition barriers.

*ACM MM, 2024, Melbourne, Australia*

© 2024 Copyright held by the owner/author(s). Publication rights licensed to ACM.
ACM ISBN 978-x-xxxx-xxxx-x/YY/MM
https://doi.org/10.1145/nnnnnnn.nnnnnnn

Compared to traditional codec methods, learning-based neural methods have demonstrated superior rate-distortion (RD) performance in the field of image compression[20, 38, 43, 47, 48]. The development of these methods falls into two main categories: (1) more powerful transform and (2) more accurate entropy estimation. The former [8, 14, 18, 31, 32] improves the compression performance by providing more efficient latent representation and usually relies on complex models with large capacity backbone. The latter directly determines the length of the codec bitstream by providing an estimation of the probability distribution. Entropy model enhances compression performance mainly by incorporating hyper-prior references, sophisticated probability models, and contextual prediction. The hyper-prior requires extra bits to save the information and focus on further removing the spatial correlation in latent representation. More sophisticated probability models like the Gaussian Mixture Model (GMM) improve the estimation accuracy by introducing more complex and large probability models to describe data distributions. The context model utilizes previously decoded data as priors to provide contextual references for undecoded data, enhancing RD performance without requiring additional bit storage for the prior information.

With the Earth as the sole observation target, RS images exhibit more consistent spatial and temporal redundancy and repetitive patterns than daily images. When applying existing computer vision (CV) methods to RS imagery in a straightforward way, the compression tends to be sub-optimal due to a lack of consideration of these characteristics. Therefore, in the field of learned RS image compression, some works incorporate edge information [16, 19] to guide compression, while others [13, 26, 44] introduce additional branches to handle different modalities of RS data to enhance the RD performance. These innovations mainly focus on adjusting and improving the main transformation part of the compression network for RS images, making it challenging to migrate these dedicated methods to the latest compression architectures. Alternatively, the context model leverages previously decoded information as priors and can be conveniently integrated into various compression frameworks based on entropy models. Although various context models achieve great progress in enhancing the reconstructed image quality and saving bitrates, two major challenges remain to be solved for RS scenarios.

Based on the above considerations, in this work, we employ the context model as our major approach, and propose methods that can better utilize the inherent features of RS images to achieve better compression. By designing methods for such more specific scenarios, we also hope to derive insights and techniques that can benefit compression method in general.

Compared to daily images, the first issue to consider is that RS images, due to patterns either formed by human beings or the nature, contain more consistent content duplication, such as intersecting farmland and sparse residential areas, or exhibit homogeneity, such

as vast expanses of bare land and forest areas. Given that commonly utilized context models rely on previously decoded elements as priors to enhance distribution estimation, expanding the receptive field during contextual referencing can help guide the entropy model to produce more precise estimations. However, existing approaches with context prior in a larger range are unable to be parallelized [25] because of the inherent limitation of auto-regressive prior[34], making the decoding process extremely time-consuming. Meanwhile, some of the parallelized methods[20, 21, 24] sacrifice the utilization of partial contextual prior, limiting the performance improvement brought by the extension of the context-aware field.

The second issue is that remote sensing images often exhibit temporal characteristics. Satellite revisits capture multiple images of the same location at different time, resulting in significant redundancy across temporal views. Though many video compression studies consider temporal inter-frame redundancy, the assumptions underlying these studies do not hold for remote sensing images. For example, the content changes in temporal RS images often manifest as abrupt transitions rather than smooth, continuous changes in videos. Therefore, exploring cross-image correlations among temporal images is also crucial in RS imagery compression models.

These limitations restrict the compression efficiency on RS data, hindering the future application of large-scale RS data. To solve these issues, we propose a spatial-temporal context model that jointly explores the correlation between spatial contents and temporal views of RS images. Specifically, we propose a stacked diagonal masked module for better spatial contextual prediction and entropy estimation. This method enables the expansion of information references by layer stacking while maintaining algorithmic parallelism. Besides, we design a mechanism incorporating latent context from different temporal images at the same geographic location within the entropy model. The temporal context reference empowers the model to leverage correlations across images, enhancing compression efficiency. Owing to the flexibility of the context model, we embed our spatial-temporal context model (STCM) into SOTA compression architectures. The experiments demonstrate that our STCM improves the performance of these architectures on RS data. When validating the quality of reconstructed images based on downstream task accuracy, our method achieves a compression ratio of 52x for single-temporal remote sensing images in scene classification without compromising accuracy. Our contributions can be summarized as follows:

- We propose a spatial-temporal context model for RS image compression, leveraging both spatial and temporal correlation to improve the compression performance.
- We propose a stacked diagonal masked module to utilize more contextual prior with larger receptive fields and maintain its parallel capability.
- The proposed STCM improves the BD-rate on RS data when compared to the SOTA methods in terms of PSNR and MS-SSIM.

## 2 RELATED WORKS

### 2.1 Learned Image Compression

Learned image compression aims to optimize the trade-off between bit-rate and distortion. Since Ballé et al.[2] introduced an end-to-end compression model by replacing the non-differentiable quantizer with an additive uniform noise, the field of learned image compression model has experienced rapid development. The learned compression model is based on approximating the distribution of discrete quantized latent representations, called the entropy model. Ballé et al.[3] further proposed an input-adaptive entropy model, which extracts the redundancy in adjacent area by introducing extra latent variables as side-information, also call hyper-priors. After that, numerous works design various entropy estimation models[17, 30, 36, 39, 46] to improve image compression performance. Many of these investigations explore different parametric models based on different distribution assumptions, including Gaussian[34], Gaussian mixture[8], and asymmetric Gaussian[11]. In this paper, we adopt the most commonly used Gaussian distribution following the previous work[20, 21, 34].

The context model, serving as an optional but effective component within the entropy estimation module, utilizes decoded contents to provide supplementary priors. It can complement hyper-priors effectively, thereby assisting the entropy model in achieving precise probability estimation[27, 34]. Inspired by Pixel-CNN[41], Minnen et al.[34] introduced the auto-regressive component into the entropy model. Later, a series of studies proved that compression performance greatly benefits from using more context. Ma et al.[33] and He et al. [20] captured local spatial and channel-wise context. Liu et al.[31] introduced 3D convolution into the context model to extract channel-wise correlations. Besides, many studies[22, 24, 25, 38] have demonstrated the effectiveness of capturing long-range content dependencies as hyper-priors or context priors via mechanisms including attention[42]. However, the quadratic computational complexity of global spatial context capturing makes it hard to be employed for large-resolution image coding.

Additionally, the models with serial dependencies along the spatial dimension significantly break the parallelism, making the decoding process extremely time-consuming. To address this issue, Minnen et al.[35] further proposed a channel-wise context model to improve the parallelism degree. He et al.[21] adopted a checkerboard pattern for the context reference, significantly improving the spatial decoding efficiency. Based on the above optimizations, the ELIC[20] model is proposed to combine channel-wise and checkerboard spatial context jointly in a parallel way. Due to the checkerboard method achieving parallelism by using contextual reference for half of the content, the utilization of contextual priors is constrained compared to the serial context model. Additionally, the checkerboard mask pattern restricts contextual information reference to odd-numbered stacking exclusively[24], and simple nonlinear stacking exhibits performance degradation in practice, making it hard to leverage a broader receptive field. To solve this issue, we propose a stacked diagonal masked module for contextual reference, which is stacked with a residual block to prevent stacking degradation, achieving a larger receptive field and maintaining the algorithmic parallelism.

### 2.2 Remote Sensing Imagery Compression

The compression of RS imagery is primarily categorized into two types: on-board and on-ground. In the context of onboard compression, the Consultative Committee for Space Data Systems recommends the standard named CCSDS 122.0-B with orthogonal

wavelet transform. On-board compression methods must consider the limitation of computational resources due to the hardware and energy consumption constraints. Some learning-based methods with reduced-complexity framework are proposed[1] recently, which integrated compression and denoising in a potentially on-orbit way. Since the on-board compression aims to achieve a good trade-off between energy consumption and compression ratio[28], the performance is satisfying enough for on-ground applications.

The most preferred traditional coding technology for on-ground compression is JPEG2000[40]. With the continuous success of deep learning technology in various RS imagery vision tasks, many works have proposed specific models tailored for compressing remote sensing images.

Based on the structure of [3], Xiang et al.[44] utilized separate branches for the low-frequency and high-frequency features of RS imagery by discrete wavelet transformation (DWT) to divide features. Han et al.[19] enhanced the compression performance of RS images by focusing on the boundary information with a VAE-based compression architecture. Guo et al.[16] presented the CENet, which incorporates an edge extractor neural network into the compression architecture to guide compression optimization by the edge-guided loss. Fu et al.[13] chose to improve the peak signal-to-noise ratio (PSNR) performance by combining transformer-based hyper-priors and CNN-based hyper-priors. Besides, GAN-based compression methods are also a trend to achieve high-fidelity compression at extremely low bit-rates[7, 37, 45]. For the consideration of efficiency, Chong et al.[9] proposed the high-order Markov random field (MRF) attention network to accelerate the convergence during training. Regarding the consideration of multi-spectral bands, Li et al.[29] utilized CNNs in conjunction with Nonnegative Tucker Decomposition (NTD) to improve reconstructed image quality and compression efficiency. Kong et al.[26] designed a multi-scale spatial-spectral attention network based on 7-band Landsat-8 and 8-band WorldView-3 satellites.

Existing literature provides valuable insight into learned RS image compression. However, few studies have investigated the temporal redundancy between temporal views of the same location. To complement existing methods, we further include the correlation extraction between temporal RS images in a context model to enhance the RD performance.

## 3 METHODS

### 3.1 Overview

Our codec is built on an entropy model with adaptive spatial-temporal contextual (STC) coding to achieve better rate-distortion performance on RS imagery. Following the previously learned image compression works, the target of our STCM is to optimize the compression ratio and the image quality metric jointly, and the overall architecture is shown in Fig 1(a).

Given an input image $x_i$ with temporal index $i$, it is transformed to latent representation $y_i$ via an encoder network $g_a$, also called analysis network in previous works. Then, the spatial dependencies of $y_i$ are captured into the hyper-priors $z_i$ via a hyper encoder $h_a$. The latent $y_i$ and hyper latent $z_i$ are quantized to $\hat{y}_i$ and $\hat{z}_i$ respectively by the quantization operator. After that, the quantized latents

$\hat{y}_i$ and $\hat{z}_i$ are compressed into bit-streams using the arithmetic encoder (AE) [12] blocks based on the mean and scale parameters given by the entropy model. The bit-streams are saved in a file as the final compressed representation of the input image $x_i$.

For decoding, the arithmetic decoder (AD) are used to decompress the saved bit-streams. In the AE and AD processes, a non-parametric entropy model is used for hyper latents $\hat{z}_i$, and the learned entropy model is responsible for providing a more accurate probabilistic estimation over the quantized latents $\hat{y}_i$. Note that the AE and AD blocks are lossless compression, and the arithmetic coding algorithm assigns fewer bits for the symbol with more occurrence frequency. Therefore, the accuracy of the entropy model directly determines the length of saved bit-streams. Following previous works [20, 34], the entropy model combines the hyper-priors and context models. For hyper-priors, the quantized hyper latents $\hat{z}_i$ are decoded with the AD first. After that, the hyper decoder $h_s$ in the hyper-prior model is applied to $\hat{z}_i$ to generate the hyper-prior $\Psi$, which will be used for distribution estimation with context prior $\Phi$ later. The hyper-prior model needs extra bits to save the hyper-prior information, while the context model utilizes the prior information from previously decoded latent. The partially decoded latents are denoted as $\langle \hat{y}_i \rangle$, which gives the context-based predictions $\Phi$. Specifically, for the single image compression, the context prior $\Phi$ contains previously decoded spatial and channel latents of the input image. For the temporal image compression, we further include the decoded latents $\hat{y}_{i-1}$ of previous image $x_{i-1}$ as temporal context prior into $\Phi$ for the distribution estimation of $x_i$. The temporal context prior is set to zero to deal with the first image in the temporal image compression. We exclude the temporal context module for single-temporal images and rely on the other prior information modules within the context model. A detailed illustration of the context model is shown in Figure 1(b) and Section 3.3. After that, the hyper-prior $\Psi$ acts as a correctness and complement for context priors $\Phi$. Here, $\Phi$ in conjunction with $\Psi$ forms the condition of the predicted Gaussian model. They are aggregated inside the parameter distribution estimator, generating the final estimated distribution parameters (mean $\mu$ and scale $\sigma$) for the conditional Gaussian entropy model. After the AD block decodes the bit-stream step by step, the decoded latent $\hat{y}_i$ is fed into the decoder $g_s$ and is transformed back to the reconstructed image $\hat{x}_i$.

The loss function for training is to minimize the expectation of rate-distortion (RD), which is defined in Equation (1).

$$\mathcal{L} = R + \lambda \cdot D$$
$$= \underbrace{\mathbb{E}_{\boldsymbol{x} \sim p_{\boldsymbol{x}}} \left[ -\log_2 p_{\hat{\boldsymbol{y}}}(\hat{\boldsymbol{y}}) \right]}_{\text{rate (latents)}} + \underbrace{\mathbb{E}_{\boldsymbol{x} \sim p_{\boldsymbol{x}}} \left[ -\log_2 p_{\hat{\boldsymbol{z}}}(\hat{\boldsymbol{z}}) \right]}_{\text{rate (hyper-latents)}} + \underbrace{\lambda \cdot \mathbb{E}_{\boldsymbol{x} \sim p_{\boldsymbol{x}}} \| \boldsymbol{x} - \hat{\boldsymbol{x}} \|_2^2}_{\text{distortion}} \quad (1)$$

where $\lambda$ is the Lagrange multiplier that determines the trade-off between the rate $R$ and the distortion $D$. $p_x$ is the distribution of the original images and $p_{\hat{*}}$ is the discrete entropy estimation model. Here, the rate term corresponds to the cross entropy between the latent marginal distribution and the learned entropy model, which will be minimized when these distributions are identical. Mean square error (MSE) is used as the distortion metric during the training stage.

Since we mainly focus on the adaptive spatial-temporal contextual model for RS imagery, we keep the network architecture of

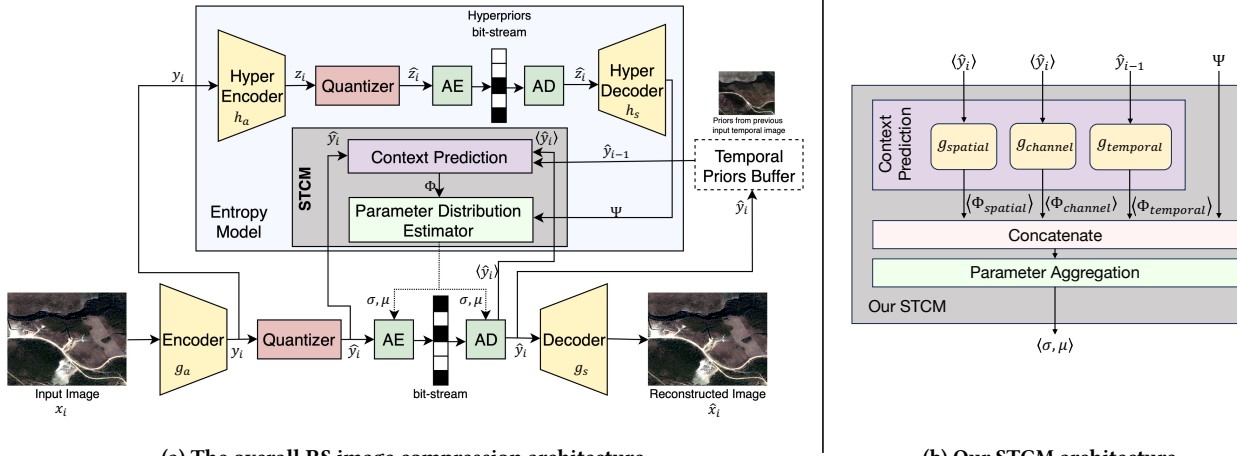

(a) The overall RS image compression architecture.

(b) Our STCM architecture.

**Figure 1: The overall RS image compression architecture of this work and the proposed Spatial-Temporal Contextual Model (STCM).**

the main transform the same as the previous works [20, 44], including the encoder, decoder, hyper-encoder, hyper-decoder, and the factorized entropy model for hyper-prior latents.

## 3.2 Spatial Contextual Usage with Stacked Diagonal Masked Module

RS imagery exhibits more content duplication than daily images since it usually contains more repeated instances in a single image. Pixels in a bigger neighboring area are likely to have stronger correlations than daily images. This implies that the larger receptive field has more chance to capture more mutual information between current decoding pixel and previously decoded ones, enabling the model to achieve better bit saving in RS images.

Figure 2 shows two commonly used mask patterns and the proposed diagonal masked pattern. The serial pattern[25, 34], shown in Figure 2(a), utilizes most contextual prior information among current learning-based context models but suffers from the serial decoding constraints (i.e., in raster scan order), making the decoding process very time-consuming and unacceptable for practice. To deploy the context model in real-world applications, the checkerboard-shaped mask strategy[20, 21], shown in Figure 2(b), enables efficient parallelism during decoding by making half of the pixels use no contextual information. However, the checkerboard method limits the neighbors used for context prediction and entropy modeling compared to the serial mask. The experimental result in Table 3 shows that the contextual information discarding in the checkerboard method limits the achievable compression ratio when satisfying the RS downstream analytic task requirements.

Therefore, we propose the diagonal pattern mask in the context model and enlarge the receptive field by stacking such masked layers with residual bottleneck blocks, which maintains the algorithmic parallelism. As shown in Figure 2(b) and Figure 2(c), we illustrate the receptive field enlargement by stacking masked layers. The checkerboard masked layer has to be stacked an odd number of times to maintain the information utilization pattern because a valid context model can only access the latents that have already

been decoded. When stacking two times, the receptive field based on checkerboard-masked convolution layers remains unexpanded because the increased reference positions have not been decoded. When stacking three layers of mask convolutional layers, the receptive field of the checkerboard expands accordingly. However, only half of the pixels can utilize the contextual information within this receptive field. Decoded pixels (blue cells) cannot use any contextual information, limiting overall contextual usage. Our proposed diagonal masked module increases the order of parallelism compared to the serial masked model by decoding the pixels diagonally in parallel from top left to bottom right and can enlarge the receptive field flexibly by enabling arbitrary times of layer stacking.

Meanwhile, although the serial mask can theoretically reference all previous latents, in practice, the context usage is limited with a single layer of $5 \times 5$ masked convolution[34]. This is because the original network based on serial mask context lacks a design to prevent degradation caused by layer stacking, which limits further bit-saving with the larger receptive field for context reference. Therefore, we introduce masked residual blocks between the masked stacked convolution layers to avoid degradation caused by layer stacking. This enables a larger receptive field and enhances the non-linearity for context modeling.

## 3.3 STCM: Spatial-Temporal Context Model

The context model leverages previously decoded information to help estimate the distribution of the current decoding area. Consequently, the compression performance enhancement brought by the context model improves with the increasing correlations between the referenced contents and the current decoding contents. RS images typically exhibit stable landscapes, resulting in strong correlations between images captured at the same location but at different times. Therefore, incorporating temporal context references in the context model can reduce temporal information redundancy, improving the RD performance of compression methods.

The context prediction in our method includes the spatial, channel, and temporal redundancy elimination, shown in Figure 1(b). We follow the previous ELIC [20] for the channel-wise context

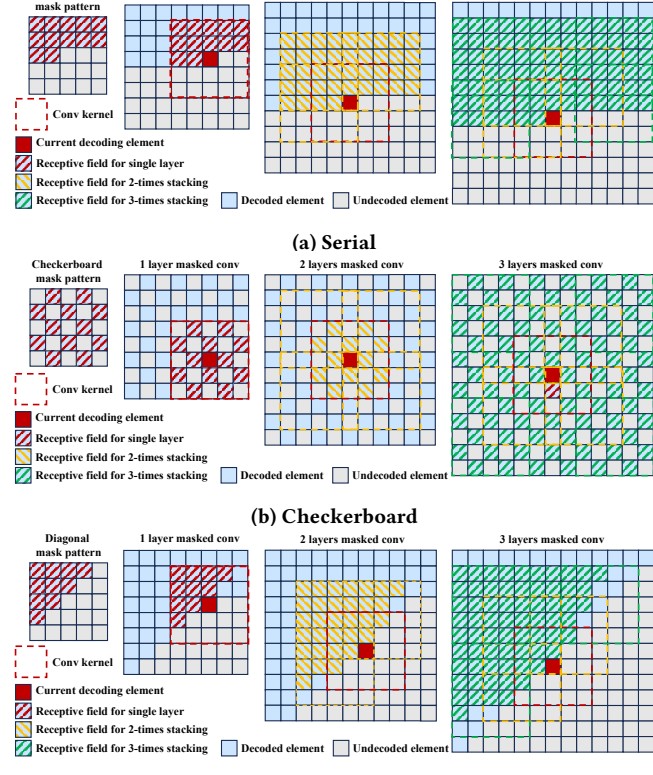

(a) Serial

(b) Checkerboard

(c) Diagonal (Ours)

**Figure 2: Receptive field comparison with different masked pattern stacking in context modeling (using 5×5 masked convolutions as example).**

model to recognize channel-wise redundancy by the uneven grouping strategy. For the spatial context model, we use the proposed stacked diagonal masked module for spatial contextual reference inside each channel group, which can be parallelized. Additionally, we introduce the previously decoded temporal image latents as temporal context for denser referring. Since the referenced temporal image has already been fully decoded before processing the current image, this temporal context can also act as a complement to the limited neighbor utilization [21, 34, 35] for entropy modeling. The output of each context branch is concatenated with the decoded hyper-prior $\Psi$. Then, all priors are fed into the final distribution estimation module to aggregate all these priors parameters, providing the entropy parameters $\mu$ and $\sigma$ of current decoding contents.

## 3.4 Inference-Stage Extension for RS imagery

Due to the variations in satellite revisiting intervals and weather conditions, the quantity of temporal RS images remains uncertain. Additionally, the varied types and quantities of sensors on satellites lead to differences in both the resolution and band numbers of RS images. This diversity makes it challenging to train models based on multi-source RS data. We cope with this issue from the inference side by training a compression model that can be used for RS image groups with varying temporal views and different band numbers.

We utilized three bands (randomly selected from multi-spectral data) for the multi-spectral, enabling the model to handle different spectral distributions. During the inference process, multi-spectral images are divided into multiple sub-images composed of three bands in the pre-processing stage. Subsequently, the model trained based on three bands is sequentially applied to these sub-images.

We only consider collaborative compression between two temporal images during training for the different number of temporal views. During the inference stage, to enhance computational efficiency, we consistently place only the first image latents of the temporal sequence into the temporal prior buffer to avoid redundant calculations. Subsequently, we sequentially input other temporal images into the model for compression and decompression.

In this way, we avoid training separate models for remote sensing images with different bands and time sequences, allowing our model to apply to data from various satellites.

## 4 EXPERIMENTAL RESULTS

### 4.1 Experimental Settings

All experiments are conducted based on an open-source library named CompressAI[4], which is widely used for developing and evaluating learning-based image compression methods.

**Datasets.** We train and evaluate the compression methods on images from fMoW-full [10], a widely used RS dataset for various tasks. The fMoW dataset consists of more than 1 million images from over 200 countries, with different number of temporal views ranging from 1 to 39. The band number in this dataset varies from 3 to 8, and the ground sample distance (GSD) resolutions are in the range from 0.3m to 3.7m. Following the dataset volume used in the previous compression framework training, we randomly select images larger than 384×384 from fMoW-full to construct two subsets, fMoW-S1 and fMoW-S2, one for training the model with spatial context model and the other for the temporal context model. The subset fMoW-S1, used for the spatial model, consists of randomly selected 11.5K single temporal images. In the meantime, we build fMow-S2 for the temporal context model, where 11.5K image pairs are randomly chosen from images with more than two temporal views. We apply the trained compression models to the UC Merced (UCM) Land Use Dataset for downstream analytic tasks to further evaluate compression quality. Specifically, we use this dataset to assess the reconstructed image quality for the scene classification task. Detailed information about these datasets used in this paper can be found in Table 1. We include more datasets and corresponding experiments in our supplementary material.

**Implementation Details.** We train our context model with main transform architectures in ELIC [20] and HL-RSCompNet [44]. For each architecture, we follow the same training settings (e.g., optimizer, learning rate, batch size, training iterations) provided in their paper, respectively, for equivalent comparisons. Due to the higher reconstruction quality requirements for RS images, we introduce a larger range of Lagrange multiplier $\lambda$ settings for more different quality presets. For the context model training with ELIC[20] transform architecture, we set $\lambda \in \{0.004, 0.15, 0.45, 0.55, 0.75\}$, achieving average bits-per-pixel (bpp) ranging from 0.12 to 9.13 and peak signal-to-noise ratio (PSNR) from 28.36dB to 46.00dB. For HL-RSCompNet architecture, we set $\lambda \in \{0.01, 0.1, 0.3, 1.5, 3.0, 10.0\}$, achieving average bpp ranging from 0.08 to 3.25 and PSNR from 27.19dB to 43.38dB. During training, we randomly crop images to

**Table 1: Datasets Details for compression models and downstream application validation.**

| Datasets | Band Num | GSD(m) | Resolution | Train/Val/Test Num | Temporal Num | Source |
|----------|----------|--------|------------|--------------------|--------------|--------|
| fMoW-S1 | 3-8 | 0.3-3.7 | >384×384 | 10000/1000/500 | 1 | WordView-2, WordView-3, QuickBird-2 |
| fMoW-S2 | 3-8 | 0.3-3.7 | >384×384 | 10000/1000/500 | 2 | WordView-2, WordView-3, QuickBird-2 |
| UCM | 3 | 0.3 | 256×256 | 2100 | 1 | Aerial |

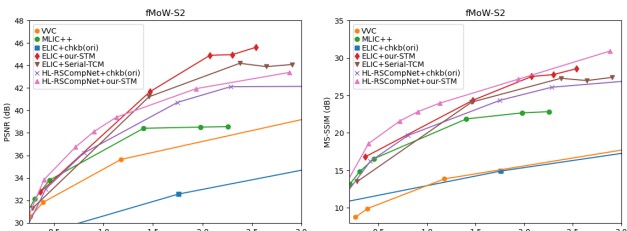

**Figure 3: Rate-distortion curve comparison for different methods based on PSRN and MS-SSIM.**

$256 \times 256$ patches, and all architectures are trained with the joint optimization of the bpp and MSE. We train our model on a single NVIDIA A800 GPU. Please refer to the supplementary material for detailed training and evaluation settings, data acquisition and pre-processing methods, and the detailed model architecture of main transform networks for each experiment.

**Evaluation.** The compression efficiency is assessed with the bpp, and the quality of reconstructed images is evaluated using the PSNR and the multi-scale mean structural similarity index measure (MS-SSIM). To evaluate the rate-distortion (RD) performance of models, we adopt the Bjøntegaard-delta rate (BD-rate) [5] as the metric, which is computed according to the bpp and a distortion metric (PSNR or MS-SSIM). For the downstream RS task, we use accuracy as the metric for scene classification.

## 4.2 Performance Comparisons

The proposed method is compared with the best traditional codec standard VTM-23.1[6] and existing learned image compression methods, including ELIC[20], MLIC++[23], and HL-RSCompNet[44]. ELIC and MLIC++ are learned image compression methods for daily images, while HL-RSCompNet is designed for RS images. We evaluate the performance improvement of our proposed context model by integrating it into two popular learning-based models designed for daily image compression (ELIC[20]) and RS image compression (HL-RSCompNet[44]), respectively. ELIC+Our-SM represents a model with the same transform architect as ELIC but embedded with our spatial context model, and ELIC+Our-STM represents the model integrated with our spatial-temporal context model. Similarly, HL-RSCompNet+Our-SM and HL-RSCompNet+Our-STM denote the models with the same main transform architecture as HL-RSCompNet with our spatial and spatial-temporal context model, respectively. We assess our models on rate-distortion performance and their impacts on the accuracy of downstream tasks. The anchor rate-distortion performance is set to VVC (VTM-23.1[1]) [6], whose BD-rate will be 0%. Besides, for different compression methods, we further compare the quality of reconstructed images by evaluating

---

[1]Official implementation: https://vcgit.hhi.fraunhofer.de/jvet/VVCSoftware_VTM/-/releases/VTM-23.1

the decreased ratio of downstream scene classification tasks based on the UC Merced Land Use Dataset.

**Rate-distortion (RD) performance.** Figure 3 shows the RD performance of our proposed spatial-temporal context model and other existing learning-based approaches on the test set of fMoW-S2. In Table 2, we report the detailed BD-rate and codec time for each method. Our context model achieves the best performance on the RS images compared with the competing methods. Our spatial-temporal context model reduces BD-rate by 61.155% with the same main transform architecture in ELIC over VTM-23.1. In contrast, the original ELIC method performs worse than VTM-23.1 on RS images, necessitating an additional 12.523% of bits to achieve similar PSNR quality. For HL-RSCompNet, which is specifically designed for remote sensing image compression, our context model further achieves an additional 11.79% bit savings. Compared with MLIC++, our temporal-context model with ELIC backbone can reduce BD-rate by up to 24.32% over MLIC++[23]. Our context model also achieves a significant BD-rate improvement in the evaluation based on MS-SSIM as the quality metric.

Figure 4 illustrates two examples of the visual comparison on the fMoW-S1 dataset. In each case, the first column denotes the original image (Ground truth). For every image, the top row displays the reconstructed image, with quantitative metrics below showing the bpp of each compressed image, MSE between the reconstructed and original images, PSNR, and MS-SSIM of the reconstructed image. To provide a more intuitive representation of reconstruction errors, the second row of each image displays the residuals between the reconstructed image and the original. Brighter areas in the residual image indicate greater information loss in those regions. Figure 4(a) shows an example with homogeneous content in a cropland area, and Figure 4(b) presents an image containing more complicated elements. Our method achieves the best results in both scenarios due to the expanded receptive field and stacked mask convolutions, which effectively extract spatial correlations. Notably, the VVC and ELIC methods suffer significant losses in boundary information, such as roads and buildings. A comparison of quantification metrics reveals that our context model effectively reduces information loss. The PSNR value of our reconstructed images is 6dB higher than the image reconstructed by VTM-23.1, and our reconstructed images retain more details with lower bpp. In terms of visual quality, our context model also provides significant improvements.

**Downstream tasks validation.** Daily image compression mainly focuses on reducing transmission and storage costs for visual applications. Remote sensing images, in addition to visual applications, are primarily used for automatic downstream tasks such as scene classification. It is essential to consider the negative impact on downstream tasks resulting from lossy compression.

We choose scene classification task for evaluation, as it is one of the most common downstream applications in RS. We train the

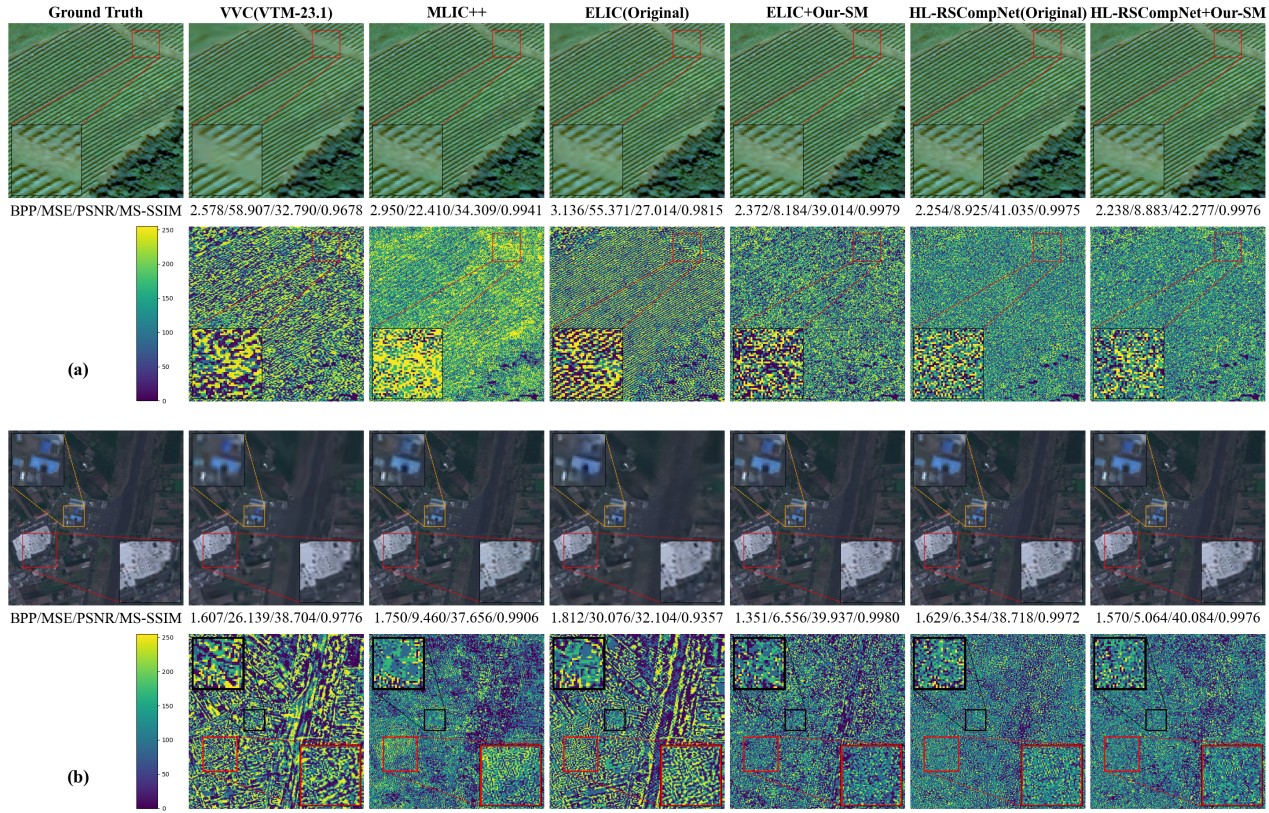

**Figure 4: Visual comparison results of two examples on fMoW-S1 dataset. The top row of each image displays the reconstructed image. To further represent reconstruction errors, the second row displays the residuals between the reconstructed image and the original image. Brighter areas in the residual image indicate more significant information loss in those regions.**

**Table 2: Comparisons of BD-Rate and codec time of different models. The Enc. and Dec. time refer to average encoding and decoding time, including arithmetic coding time. All evaluated methods take VVC as an anchor to calculate the BD-rate.**

| Model | BD-Rate for PSNR(%)↓ | BD-Rate for MS-SSIM(%)↓ | Enc. Time(s) | Dec. Time(s) |
|---|---|---|---|---|
| VVC(VTM-23.1) | 0 | 0 | 6.482 | 0.056 |
| MLIC++[23] | -36.834 | -78.940 | 0.089 | 0.129 |
| ELIC+Chkb-CM(Original) | 12.523 | -58.762 | 3.41 | 3.001 |
| ELIC+Our-SM | -55.831 | -81.917 | 1.892 | 2.010 |
| ELIC+Our-STM | -61.155 | -84.893 | 3.353 | 2.971 |
| HL-RSCompNet+Chkb-CM(Original) | -40.682 | -81.034 | 0.038 | 0.014 |
| HL-RSCompNet+Our-SM | -44.076 | -81.654 | 0.032 | 0.571 |
| HL-RSCompNet+Our-STM | -52.480 | -87.999 | 0.034 | 0.617 |

MSMatch [15] network on the UC Merced Land dataset, which is one of the widely accepted SOTA methods. We follow all MSMatch settings to acquire the trained classification model.

To compare the effects of compressed images on downstream tasks, we conduct inference with the trained classification model on both the original images in the UCM test set and the reconstructed images using different compression models. The results are shown in Table 3. Our method achieves the lowest bitrate with

a compression ratio of 52x without compromising scene classification accuracy. In comparison, the ELIC method requires over 20 times more bits than ours to achieve similar results, while MLIC++ requires nearly four times the bits of our method.

We provide additional results and analysis on other downstream tasks and datasets in our supplementary material.

**Table 3: Downstream scene classification task accuracy comparisons based on reconstructed images from various methods and quality settings (i.e., bpp). The highlighted blue data indicates that the accuracy of the downstream tasks remain unaffected.**

|     | No Comp | VVC | | | ELIC(Ori) | | | MLIC++ | | | Ours+HL_RSCompNet | | | Ours+ELIC | | |
| --- | --- | --- | --- | --- | --- | --- | --- | --- | --- | --- | --- | --- | --- | --- | --- | --- |
| bpp | - | 0.49 | 1.43 | 4.18 | 0.17 | 2.18 | 9.51 | 0.26 | 0.58 | 1.81 | 0.1 | 0.56 | 1.08 | 0.46 | 2.25 | 3.12 |
| Acc | 0.991 | 0.705 | 0.891 | 0.952 | 0.871 | 0.981 | 0.991 | 0.976 | 0.981 | 0.991 | 0.895 | 0.991 | 0.991 | 0.991 | 0.991 | 0.991 |

**Table 4: Ablation study of spatial context model on fMoW-S1.**

|     | Context Model | | | Layer Num | | Stack | BD-rate |
| --- | --- | --- | --- | --- | --- | --- | --- |
|     | Serial | Chkb | Diag | 1layer | 3layers | Res block | (%) |
| (a) |     |     | ✔ |     | ✔ | ✔ | 0 |
| (b) |     | ✔ |     |     | ✔ | ✔ | +6.889 |
| (c) |     |     | ✔ | ✔ |     | - | +21.183 |
| (d) |     | ✔ |     | ✔ |     | - | +177.659 |
| (e) | ✔ |     |     | ✔ |     | - | +7.908 |
| (f) |     |     | ✔ |     | ✔ | ✗ | +597.104 |

## 4.3 Ablation Study

We conduct ablation experiments on the mask pattern and stacking manner of the context model. All experiments are based on the ELIC main transform architecture for fair comparison. Table 4 shows the results evaluated on the fMoW-S1 test set. Anchor performance, denoted as experiment (a) in Table 4, is set to our stacked diagonal masked context model, which is stacked three times with residual block. The increase in BD-rate implies the additional bits required to achieve comparable image quality (i.e., PSNR).

For the masked pattern in the context model, we evaluate the impact of three patterns on the performance of the compression model, including serial (Figure 2(a)), checkerboard (Figure 2(b)), and diagonal (Figure 2(c)) patterns. The results of experiments (c), (d), and (e) demonstrate that, when all patterns use a single-layer masked convolution, the serial mask pattern outperforms the diagonal pattern, which in turn surpasses the checkerboard pattern. The reason is that the checkerboard mode discards half of the spatial context utilization by skipping every other position during the decoding process, while for serial and diagonal modes, all pixels utilize spatial context information during decoding. Specifically, serial mode utilizes slightly more spatial context during each convolution than diagonal mode. All these results suggest that spatial contextual information utilization plays a crucial role in accurately reconstructing intricate details within the RS images.

Additionally, we conduct experiments on different stacking manners of the context model, comparing the single-layer context model, the simple nonlinear three-times stacking context model, and our three-times stacking context model with residual blocks. According to the experimental results comparisons between experiments (a) and (c), as well as between experiments (b) and (d) in Table 4, it can be concluded that incorporating residual blocks in the contextual layer stacking enhances the compression performance of both checkerboard and diagonal masked convolution. Besides, simply stacking with nonlinear layers has a significantly detrimental impact on compression performance, as evidenced in the results of experiments (a) and (f). This underscores the effectiveness of residual blocks in mitigating performance degradation induced

by convolution layer stacking in the context model. We excluded stacking experiments for serial masked convolution due to its serial dependencies. The inability for parallel decoding processes results in significant time overheads for the algorithm. Stacking would substantially prolong its decoding time, severely limiting the practical applicability of the algorithm.

## 4.4 Discussion of Critical Factors for RS Image Compression

We discuss critical factors influencing RS image compression. Firstly, expanding the receptive field is crucial for acquiring more relevant contextual information for RS images. As shown in Table 2 and Table 4(a-b), methods with larger receptive fields, whether achieved by attention mechanisms like MLIC++ or convolution stacking like experiments (a) and (b) in Table 4, can achieve better compression performance on RS data. Secondly, the quantity of contextual references plays a vital role in RS image compression. Results in Table 2 and Table 4(e) demonstrate that the serial method and our context model method outperform the original checkerboard method. The result confirms that the effectiveness of the checkerboard-shaped contextual mask is limited for RS images, as it discards half of the contextual reference. Thirdly, separating high and low-frequency information enhances the performance of RS image compression. HL-RSCompNet performs well on RS data, even though the main transform network of this model is relatively simple and lightweight. We attribute this to the strategy proposed in this method, which deals with high and low-frequency information separately. This helps the model integrate multimodal information from different sensors, and the high-frequency information process branch can assist the preserving of more critical details in RS images.

## 5 CONCLUSION

This paper proposes a spatial-temporal context model (STCM) for remote sensing imagery compression, which can take advantage of spatial and temporal redundancies in RS imagery jointly. Based on our STCM, we further improve the performance of the state-of-the-art models for RS imagery. The significance of our work lies in leveraging the characteristics of redundancy for RS data within the context model, which enhances the performance of compression methods on RS images. We validate the compression models by assessing their impacts on RS downstream tasks, such as scene classification. This evaluation helps ensure the usability of compressed data in practical scenarios. Our STCM achieves up to 52x compression ratio for single temporal images without decreasing the accuracy of the scene classification task. Our compression model reduces the data storage and retrieval costs associated with such applications, thereby enabling their scalability to larger scales.

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
