# OpenReview forum: "Spatial-Temporal Context Model for Remote Sensing Imagery Compression"
_acmmm.org/ACMMM/2024/Conference — MM2024 Poster_

### Official Review · Reviewer_VdHi · 2024-04-30

**Rating:** 4
**Confidence:** 2

**Summary:**

The paper presents a novel Spatial-Temporal Context Model (STCM) for compressing remote sensing (RS) imagery, addressing the challenges posed by the high spatial and temporal resolutions of RS images. The authors highlight that traditional image compression methods do not fully exploit the unique characteristics of RS images, such as content duplication, homogeneity, and temporal redundancy. To overcome these limitations, the proposed STCM leverages context from both spatial and temporal perspectives to enhance compression efficiency.

**Strengths:**

* The paper is technically rigorous, with a clear explanation of the mathematical models and algorithms used, which suggests a high level of technical correctness.

* The paper is well-structured, with a clear presentation of the problem, the proposed solution, and the results. The figures and tables are informative and contribute to a better understanding of the content.

* The paper discusses the application of the proposed model to RS images from various sources, demonstrating its versatility and potential for real-world use.

**Limitations:**

* The article appears to simply apply generic learned image compression methods to remote sensing image datasets without specifically illustrating the uniqueness of the proposed approach on remote sensing datasets in the experimental section.

* The article presents a comparison with only four methods, where VTM-23.1 is an older approach, and both ELIC and MLIC++ are designed for general image compression, with only HL-RSCompNet being specifically tailored for remote sensing imagery. The lack of inclusion of more recent and representative methods in the following significantly diminishes the persuasiveness of the article's findings.
[1] Learned Distributed Image Compression with Multi-Scale Patch Matching in Feature Domain. AAAI 2023: 4322-4329
[2] Learned Image Compression with Mixed Transformer-CNN Architectures. CVPR 2023: 14388-14397
[3] LVQAC: Lattice Vector Quantization Coupled with Spatially Adaptive Companding for Efficient Learned Image Compression. CVPR 2023: 10239-10248
[4] Learned Lossless Image Compression Through Interpolation With Low Complexity. IEEE Trans. Circuits Syst. Video Technol. 33(12): 7832-7841 (2023)
[5] Revisiting Learned Image Compression With Statistical Measurement of Latent Representations. IEEE Trans. Circuits Syst. Video Technol. 34(4): 2891-2907 (2024)

* The  spatial-temporal context model (STCM)  is main contribution of the paper, however, in the  experiments section, the effectiveness of the component is not validated.

* Regarding to the table 2, the Dec. Time for HL-RSCompNet+Chkb-CM(Original) is 0.014, when embedding with the proposed SM and STM, the Dec. Time increases to 0.571 and 0.617, which makes me worry about the execution efficiency of the algorithm.

*  The paper mentions  "a compression ratio of 52x ", the correct expression for x should be \times $\times$

**Suitability:**

2

---

### Official Review · Reviewer_QNnm · 2024-05-24

**Rating:** 3
**Confidence:** 2

**Summary:**

This paper proposes a Spatial-Temporal Context model (STCM) for RS image compression, jointly leveraging context from a broader spatial scope and across different temporal images.

**Strengths:**

1. This paper explains the structure of the method very clearly, and the drawing is also very fine, which can intuitively understand the proposed method.
2. Compared with some algorithms, the proposed algorithm achieves better results.

**Limitations:**

1. There are a few experimental models and more tests need to be added.
2. Regarding whether Stacked Diagonal Masked Module is better, the explanation in the paper does not convince me. There are no corresponding experiments, such as visual diagrams, to prove its effect.

**Suitability:**

2

---

### Official Review · Reviewer_oKCr · 2024-05-25

**Rating:** 4
**Confidence:** 2

**Summary:**

In this manuscript, a Spatial-Temporal Context model (STCM) is proposed for RS image compression that jointly utilizes contexts from a wider spatial range and across different temporal images. The authors extend the range of context references by a stacked diagonal mask module, which is stackable and maintains its parallel capability. In addition, the authors propose spatial-temporal contextual adaptive coding, which enables entropy estimation to reference contexts from different temporal RS images of the same geographical location.

**Strengths:**

1. The proposed method achieves state-of-the-art results in the RS image compression domain.
2. Remote sensing images from different temporal characteristics are fully utilized.
3. The authors think deeply about the key factors that affect RS image compression.

**Limitations:**

1. On page 3, lines 270-272, the authors state that to complement the existing approach, they further include correlation extraction between temporal RS images in the contextual model to improve RD performance. However, in Table 1, two of the datasets used by the authors have a temporal quantity of 1. It is suggested that the authors add another multi-temporal dataset to demonstrate that the model can fully utilize the correlation between temporal RS images.
2. In Figure 4 (b), the difference between HL-RSCompNet (Original) and HL-RSCompNet+Our-SM is not obvious and needs to be explained by the authors.
3. The comparative experiments in Table 2 are relatively few, and the authors need to add more comparative experiments to further validate the advantages of the proposed method.
4. In Table 4, the authors need to add the time overhead to further demonstrate the effectiveness of the proposed method.
5. What are the advantages of the authors' proposed method over tensor compression?

**Suitability:**

2

---

### Meta-Review · Area_Chair_h8v2 · 2024-07-04

**Recommendation:** Accept (Poster)
**Confidence:** 4

**Metareview:**

The paper introduces a novel Spatial-Temporal Context Model (STCM) designed specifically for remote sensing (RS) image compression. It leverages the unique properties of RS images, such as content duplication, homogeneity, and temporal redundancy, to improve compression efficiency. The proposed model incorporates a stacked diagonal masked module to expand the contextual reference scope while maintaining parallel processing capabilities. Additionally, spatial-temporal contextual adaptive coding is introduced to reference context across different temporal images at the same geographic location, enhancing entropy estimation. The method shows superior rate-distortion (RD) performance compared to existing compression techniques and significantly reduces the bit rate for downstream tasks like scene classification, maintaining high accuracy.

From reviewers and overall assessment I recommend the paper for acceptance (poster).